# Investigation of MAF for Finishing the Inner Wall of Super-Slim Cardiovascular Stents Tube

**DOI:** 10.3390/ma16083022

**Published:** 2023-04-11

**Authors:** Guangxin Liu, Yugang Zhao, Zhihao Li, Chen Cao, Jianbing Meng, Hanlin Yu, Haiyun Zhang

**Affiliations:** School of Mechanical Engineering, Shandong University of Technology, Zibo 255049, China

**Keywords:** CBN magnetic abrasive, MAF, cardiovascular stents, Ni-Ti alloy, response surface optimization

## Abstract

The internal wall of cardiovascular stent tubing produced by a drawing process has defects such as pits and bumps, making the surface rough and unusable. In this research, the challenge of finishing the inner wall of a super-slim cardiovascular stent tube was solved by magnetic abrasive finishing. Firstly, a spherical CBN magnetic abrasive was prepared by a new method, plasma molten metal powders bonding with hard abrasives; then, a magnetic abrasive finishing device was developed to remove the defect layer from the inner wall of ultrafine long cardiovascular stent tubing; finally, response surface tests were performed and parameters were optimized. The results show that the prepared spherical CBN magnetic abrasive has a perfect spherical appearance; the sharp cutting edges cover the surface layer of the iron matrix; the developed magnetic abrasive finishing device for a ultrafine long cardiovascular stent tube meets the processing requirements; the process parameters are optimized by the established regression model; and the inner wall roughness (Ra) of the nickel–titanium alloy cardiovascular stents tube is reduced from 0.356 μm to 0.083 μm, with an error of 4.3% from the predicted value. Magnetic abrasive finishing effectively removed the inner wall defect layer and reduced the roughness, and this solution provides a reference for polishing the inner wall of ultrafine long tubes.

## 1. Introduction

Over the past few decades, cardiovascular diseases (CADs) have become the leading cause of human health, causing tens of millions of deaths each year [1,2,3]. Among the many methods of treatment for cardiovascular diseases such as angioplasty, drugs, and bypass grafting, the vascular stenting technique stands out as the main treatment for these diseases due to its unique advantages of immediate patency, minimally invasive nature, and low risk of restenosis [4,5,6,7,8,9]. Nickel–titanium (Ni-Ti) alloys are not only memory alloys but also have super elasticity, fatigue resistance, corrosion resistance, and high mechanical properties, and are widely used in the manufacture of cardiovascular stents [10,11,12]. Ni-Ti cardiovascular stents have developed defects such as pits, bulges, and cracks in the inner wall during the drawing production process. The presence of these defects puts the patient’s health at risk, and the inner wall of cardiovascular stent tubing must have a low roughness, which is important for improving the fatigue resistance and cytocompatibility of medical implants [13,14,15].

Current research on the polishing of the inner walls of cardiovascular stents is focused on the direction of chemical polishing. Brecht van Hooreweder et al. treated cobalt–chromium F75 alloy stents produced by SLM using a mixture of hydrochloric acid and hydrogen peroxide, and showed that a 27% hydrochloric acid and 8% hydrogen peroxide solution was effective in removing adherent particles and maintaining the fatigue resistance of the stents [15]. Wojciech Simka et al. evaluated the conditions affecting the surface quality of Ni-Ti alloys by the AFM and SEM methods. The results of the study showed that after electrochemical polishing, the workpiece indicated the formation of a TiO_2_ layer, which increased the corrosion resistance of Ni-Ti alloys [16]. Li et al. used perchloric acid–glacial acetic acid for the electrolytic polishing of Ni-Ti alloy cardiovascular stent tubing [17]. Liu obtained the best combination of electrochemical polishing process parameters—a polishing voltage of 17 V, an interpolar distance of 20 mm, a polishing time of 10 min, and a polishing temperature of 25 °C—through orthogonal tests, in which the polishing voltage had the greatest influence on the polishing quality [18]. Sun et al. used electrochemical polishing of Ni-Ti alloy cardiovascular stents using an electrolyte composition of ethylene glycol–sodium chloride nontoxic electrolyte, and the experimental results showed that the surface integrity and biocompatibility of Ni-Ti alloy cardiovascular stents fabricated by this process were significantly improved, and a titanium dioxide protective film formed on the surface [19]. Huang used a self-designed electrolytic polishing device to electrolytically polish nickel–titanium alloy cardiovascular brackets; the polishing solution used was perchloric acid–glacial acetic acid, with triethanolamine and anhydrous ethanol as additives; the test results for the best surface roughness value, *Ra* = 53.8 nm [20]. Xu used magnetic field-assisted electrochemical processing to effectively improve the efficiency and quality of electrochemical polishing of nickel–titanium alloy surfaces, effectively improving the surface roughness, microscopic morphology, mechanical properties, corrosion resistance, and biocompatibility of nickel–titanium alloy [21]. Yan compared two polishing solutions—methanol–perchloric acid and glacial acetic acid–perchloric acid—in performing electrochemical polishing, and found that the latter had a better polishing effect, and the experimental results showed that the current density had the most significant polishing effect, and the optimal surface roughness obtained was 66.7 nm [22]. In the process of chemical polishing of Ni-Ti alloy, cardiovascular brackets need to control several process parameters; the process is complex, and new defects such as pitting and bulging appear after polishing, and the resulting waste solution disposal is also not standardized, causing environmental pollution. However, chemical polishing is even more helpless when faced with the polishing of ultraslender cardiovascular stent tubing, which requires larger containers to hold the chemical solution and more complex process parameters to be adjusted. Improper disposal of the used waste solution also causes environmental pollution and is not in line with green sustainability.

Magnetic abrasive finishing (MAF) is a processing method with the advantages of flexibility, low temperature rise, high efficiency, and no tool wear compensation, and the magnetic abrasive particles (MAPs) used consist of ferromagnetic metals and hard abrasives [23,24]. Currently, MAF has been applied in the medical, military, aerospace, and communication fields for the study of flat surfaces [25,26], internal and external cylindrical surfaces [27,28,29], curved surfaces [30,31], and deburring [32,33], etc. MAPs play a central role in MAF. In the past few decades, a wide range of methods for the preparation of MAPs have been developed by scholars, such as bonding [34], mechanical mixing [35,36], hot press sintering [37], plasma spraying [38,39,40], electroless plating [41], and in situ alloy hardening [42]. Due to the limitations of irregular shape and the short lifetime of existing magnetic abrasives (MAPs), only tubes with large inner diameters and short lengths can be processed at present [43,44,45,46], and they are helpless in the face of polished ultraslender cardiovascular stent tubes.

In this research, spherical CBN magnetic abrasives were successfully prepared using a method consisting of plasma molten metal powder bonding with hard abrasives, which combines the advantages of both the atomization fast coagulation method and the plasma spraying method, and has cleverly solved the disadvantages of both. The magnetic abrasives prepared by this method are characterized by high sphericity, narrow particle size distribution, long service life, and economy, which facilitate the introduction of magnetic particle grinding inside ultrafine long cardiovascular stent tubing. In this paper, processing tests were conducted using the developed magnetic particle finishing processing equipment for the inner wall of vascular stent tubing, and the process parameters of tubing speed, magnetic pole feed rate, and magnetic abrasive filling amount were optimized using the response surface method to obtain the optimal combination of process parameters to achieve the efficient removal of the defect layer from the inner wall of ultrafine long vascular stent tubing.

## 2. Spherical CBN Magnetic Abrasive Preparation Experiment

### 2.1. Principle of Magnetic Abrasive Preparation

Figure 1 shows the principal diagram of magnetic abrasive preparation by plasma molten metal powder bonding with hard abrasives. The equipment is modified from plasma powder spheroidization experimental equipment, which consists of a plasma generator, powder mixer, hard abrasive nozzle, magnetic abrasive synthesis condensation chamber, power supply and air supply vacuum system, powder collection and dust removal system, and other parts. The preparation principle is as follows: The plasma is used to heat and melt the spherical metal powder particles with narrow particle size distribution, and in the process of metal microdroplets falling, the airflow containing CBN hard abrasive is sprayed on them, so that the CBN hard abrasive is shot into the metal microdroplets, and before the CBN hard abrasive powder escapes from the metal microdroplets, the particle size is evenly distributed, and the hard abrasive is combined with the metal matrix by rapid condensation. The magnetic abrasive is firmly bonded to the metal matrix, with regular spherical shape and strong grinding and polishing performance. Due to the low heating temperature and short heating time of the CBN hard abrasive, the cutting edge of the hard abrasive is not blunted and remains intact after the magnetic abrasive is formed, thus producing a high-performance magnetic abrasive.

The reasons for preparing magnetic abrasives with narrow particle size distribution are as follows: firstly, the raw material of the iron matrix used is a spherical powder with narrow particle size distribution, and secondly, the speed of spraying hard abrasives is just enough to make the hard abrasives enter the shallow surface layer of the molten iron matrix without breaking it. These two points are the main reasons to ensure that the prepared magnetic abrasives have a narrow particle size distribution. It also makes it possible to reduce the waste of hard abrasives during the preparation of high-performance magnetic abrasives, and thus increase the yield. Since the superficial layer of hard abrasive in the magnetic abrasive is firmly fixed by the cold shrinkage of the molten iron matrix, this will make the unmelted iron matrix not be bonded with hard abrasive powder.

### 2.2. Magnetic Abrasive Preparation Experiment

In the process of preparing magnetic abrasives by plasma molten metal powder bonding with hard abrasives, in order to reduce the agglomeration phenomenon between spherical iron powder due to interdependence, spherical iron powders with Fe content >99% and particle size distribution of 180~212 μm were used without adsorption to each other. The CBN hard abrasive powder used was D_50_ = 14 μm; Table 1 shows the process parameters for the preparation of CBN magnetic abrasives.

## 3. Spherical CBN Magnetic Abrasives Characterization

### Microscopic Morphology of Spherical CBN Magnetic Abrasives

Figure 2 shows the SEM image of the prepared CBN powder. The SEM image of the CBN magnetic abrasive is produced by this method. From Figure 2, it can be seen that the CBN powder is uniformly and tightly aggregated in the superficial layer of the spherical iron matrix without agglomeration; the magnetic abrasives are spherical in shape. This excellent characteristic of the spherical shape of CBN magnetic abrasives is can easily be introduced into the interior of the ultralong cardiovascular support tube, which is mainly spherical in shape. Its appearance gives it a certain mobility, and the formation of magnetic abrasive brushes facilitates the turnover of worn and unworn magnetic abrasives. As can be seen in Figure 2, one part of the CBN particles has been firmly fixed by the iron matrix, while the other part remains exposed with sharp edges. This provides sufficient cutting performance while ensuring the longevity of the magnetic abrasive. The superficial layer of hard abrasive on the magnetic abrasive is not dulled by the high temperature, mainly due to the unique feature of the method of plasma molten metal powder bonding with hard abrasives. In this method, only the iron matrix powder is heated, not the mixture of iron matrix and hard abrasive, and the fact that the hard abrasive powder is ejected from an area away from the high temperature is another reason why the magnetic abrasive retains its sharp cutting edge.

## 4. Cardiovascular Stent Tube Inner Wall Defect Layer Removal Test

### 4.1. Principle and Equipment for Magnetic Particle Lithography of the Inner Wall of Vascular Stents

Figure 3a shows the developed equipment for removing the inner wall defect layer of ultraslender cardiovascular stent tubing by magnetic particle grinding. The equipment consists of an AC servo motor, magnetic device, synchronous belt, stepper motor, CN system, and so on. Among them, the AC servo motor has a precision chuck. The working principle of the equipment is as follows: The AC servo motor at both ends has the same speed and opposite steering to drive the ultraslender cardiovascular support rotation; at the same time, the magnetic device drives the magnetic abrasive inside the tube to perform a reciprocating motion. Under the combined effect of rotation and reciprocating motion, the magnetic abrasive inside the tube moves in a spatial spiral. The magnetic pole generates a certain pressure on the tube wall by attracting the magnetic abrasive inside the tube to remove the defect layer.

Figure 3c shows the principle of magnetic abrasive processing of the inner wall of a Ni-Ti alloy vascular stent tube. The iron-based CBN magnetic abrasive is added to the inside of the Ni-Ti vascular stent tubing, and under the action of the external magnetic field, the magnetic abrasive grains are magnetized by the external magnetic field to form a magnetic grain brush with certain cutting ability and rigidity. The nitinol vascular stent tubing is clamped by two end collets, which are mounted on servo motors at both ends. When the servo motor drives the tube to rotate and the magnetic pole moves reciprocally in the direction of the axis, the magnetic grain brush moves relative to the inner wall surface of the tube, and the magnetic abrasive, which is magnetized to form the “abrasive brush”, moves spirally on the surface of the workpiece to produce the effect of sliding, cutting and plowing on the inner wall of the tube, and then the inner wall is magnetically ground.

### 4.2. Experimental Cardiovascular Stent Tube

The length of the cardiovascular stent tubing for the test was 1800–2000 mm, with an inner diameter of 1.5 mm and an outer diameter of 2.0 mm, as shown in Figure 4, with the elemental composition shown in Table 2 and the performance parameters of the nickel–titanium alloy shown in Table 3, and the tubing of this material was not ferromagnetic.

## 5. Response Surface Method Process Parameter Optimization

### 5.1. Experimental Design and Results

The response surface method (RSM) is a mathematical method for optimization processes that integrates optimization design and statistical analysis [47]. By combining the scheme with the experiment, the response values of each group of parameters are obtained, and the response surface model between the variables and the response values is constructed to establish the functional relationship between the response target and the design variables, and the optimal process parameters are obtained by analyzing the functional relationship.

The Box–Behnken experimental design method was used for response surface establishment and analysis [47], and the BBD experimental design is a spherical design in which all points of the experimental design are at the midpoints on each side of the positive cube and at the center of the body. That is, all design points are on the sphere, spherical domain, and approximate rotatability, and trials located at the vertices of the cube are not included, which is suitable for three-level trials and can predict all main effects, so the number of BBD trials is lower, and the results obtained can have a greater range of evaluation. This test is a three-factor, three-level test with 17 sets of trials. Tube rotation velocity *A*, magnet feed velocity *B*, and MAPs filling quantity *C* were used as process parameter variables and surface roughness *Ra* was used as response value. Table 4 shows the process parameters and levels for the BBD test.

In order to investigate the optimal process parameters to obtain the magnetic abrasive finishing processing of this tube, the tube was intercepted into 300 mm before the test, thus saving the test cost. The processing time for each test was 300 min, and magnetic abrasives with a particle size of 150 μm were selected as finishing tools. The surface roughness *Ra* of five areas of the inner wall of the tube was measured using a 3D digital microscope, and an average value of 0.356 μm was taken.

### 5.2. Surface Roughness Regression Model and Its Analysis

In the process of MAF, the relationship between process parameters and the measure of surface roughness is highly complex and nonlinear, and cannot be expressed in an expression, so a response surface model is needed to represent the relationship instead of the original model. However, for most response surface design problems, the expression of the function between the independent and response variables is unknown. In order to obtain the optimal solution, it is necessary to create a suitable mathematical model from a large amount of experimental data, then express this mathematical model in graphical form, and finally determine the optimal constraint or optimal response region directly by analysis. The response surface model is based on least-squares regression for data fitting and is the most effective and simple alternative model. The mathematical expression of the polynomial response surface method is shown in Equation (1) [48,49,50,51].
(1)$$y=\beta_{0}+\sum_{i=1}^{n}\beta_{i}x_{i}+\sum_{i=1}^{n}\beta_{ii}x_{i}^{2}+\sum_{i=1}^{n-1}\sum_{j=i+1}^{n}\beta_{ij}x_{i}x_{j}+\zeta$$
where:*y* is the dependent variable;i=1,2,3,…,n;$x_{i}$ and $x_{j}$ are the design variables;n is the number of design variables;$\beta_{0},\;\;\beta_{i},\;\beta_{ii},\;\;\beta_{ij}$ are response surface regression coefficients;ζ is the fitting error.

In practical engineering applications, many response surface design problems can be approximated by the above second-order polynomial expressions, but in the process of seeking optimal solutions, a response surface model cannot be completely true to express the true function of the independent variable in the whole space, and if the response surface model can be used as a reasonable approximate expression of the true function, or if the response variable in a certain region has a good response, then the response surface model can be said to be reliable [52]. Converting the process parameters in the test, the free variables of each test factor and the surface roughness were converted into matrix form, and then the regression coefficients were obtained by the least-squares method to obtain the multiple regression equations of the surface roughness with the tube rotation velocity, magnet feed velocity, and MAPs filling quantity, as shown in Equation (2).
(2)$$Ra=0.1+0.019A+0.011B\;-\;0.01C+0.00425AB+0.00825AC+0.012BC+0.072A^{2}+0.052B^{2}+0.0002C^{2}$$

In the analysis using the ANOVA method, the *p*-value was used to determine the degree of overall influence of the change in the factors, and thus the significant influence parameters [53]. Based on the ANOVA method, the *p*-values for different structural parameters were obtained based on the constructed response surface prediction model for surface roughness, as shown in Table 5. The F-values represent the significance of the overall regression equation model and P represents the significance level of the regression equation model. As seen in Table 6, the *p*-value of the model is less than 0.0001 for the single parameter, indicating that the regression equation established between the model surface roughness and the respective variables is extremely significant, and the misfit F-value is 1.84, indicating that it is not significant. The *p*-values of 0.0005 and 0.0092 for the tube speed, pole feed rate, and abrasive filling, respectively, are less than 0.05, indicating that the effect of these three parameters on the surface roughness *Ra* is significant. The most influential one is tube rotation velocity, followed by the magnet feed velocity, and finally, the MAPs filling quantity.

The multivariate second-order prediction model of the magnetic particle-grinding process parameters and the response target obtained based on RSM can be used as the prediction model for *Ra*. The theoretical test uses the coefficient of determination *R*^2^ and the modified coefficient of determination *R*^2^*_Adj_* as indicators to evaluate the accuracy of the model, and the closer their values are to 1, the higher the accuracy of the response surface model. The expressions of the coefficient of determination *R*^2^ and the modified coefficient of determination *R*^2^*_Adj_* are as follows:(3)$$\left\{\begin{array}{c} R^{2}=\frac{\sum_{i=1}^{n}{\left(\hat{y_{i}}-\bar{y_{0}}\right)}^{2}}{\sum_{i=1}^{n}{\left(y_{0}-\bar{y_{0}}\right)}^{2}}\\ R_{Adj}^{2}=\frac{\sum_{i=1}^{n}{\left(\hat{y_{i}}-\bar{y_{0}}\right)}^{2}\left(n-1\right)}{\sum_{i=1}^{n}{\left(y_{0}-\bar{y_{0}}\right)}^{2}\left(n-p-1\right)} \end{array}\right.$$
where:n is the number of actual observations;*i* = 1, 2,..., *n*;*p* is the model degrees of freedom;$y_{0}$ is the value of the response obtained through the actual model;$y_{i}$ is the actual observed value of the *i*th response;$\hat{y_{i}}$ is the predicted value of the response surface model;$\bar{y_{0}}$ is the mean value of the actual observation.

The resultant multivariate correlation coefficient *R*^2^ is 0.9868, according to Equation (3), and the corrected multivariate correlation coefficient *R*^2^*_Adj_* is 0.9699, indicating that the regression model for surface roughness explains 96.99% of the response values. This indicates that the model is highly accurate and has a high degree of confidence. Therefore, the developed response surface model is able to replace the finite element model for predicting and analyzing the surface roughness of nickel–titanium alloy vascular stents tube after processing.

In order to accurately predict the surface roughness of the inner wall of the processed vascular stents tube, the accuracy of the response surface must be checked after the second-order response surface is established, which is divided into two steps: Step 1 is the fitness plot test; Step 2 is the theoretical test. The fitting effect of the model can be evaluated by several diagnostic plots, including the normal probability of residuals, residuals and predicted values, and predicted values and actual values, as shown in Figure 5, Figure 6 and Figure 7, where the data points of different colors represent the surface roughness *Ra* obtained under different process parameters. The residuals follow a normal distribution and the surface roughness model has good adaptability. The upper and lower lines in Figure 6 represent the distribution range of the residuals. In the absence of constant error, the residuals of all the predicted values of surface roughness *Ra* are randomly distributed in the range of ±4.82 on the near-zero axis, which indicates that there is no obvious pattern among the residuals of surface roughness Ra. A small number of data points in Figure 5 are distributed on the straight line (*y* = *x*), and most of the data points are distributed around the straight line (*y* = *x*), indicating that the experimental values of surface roughness *Ra* are in better agreement with the predicted values.

### 5.3. Response Surface Analysis

Based on the secondary response surface mathematical model of tube rotation velocity, magnet feed velocity, and MAPs filling quantity, controlling a single response factor constant, the influence of the remaining response factors on the surface roughness is studied, and the shape of contour lines and the density of curves are used to judge the size of the interaction effect of each factor on the response surface.

#### 5.3.1. Interaction between the Tube Rotation Velocity and the Magnet Feed Velocity

Figure 8 shows the contour lines and three-dimensional response surface of the interaction effect of the tube rotation speed and magnet feed velocity on surface roughness for a MAPS filling quantity of 0.15 g. As can be seen from the 3D response surface, the plot shows an ideal parabolic shape, indicating a clear interaction. The surface roughness values obtained tend to increase with increasing tube rotation velocity and increasing magnet feed velocity, while too small a magnet feed velocity and too slow a tube rotation velocity are not conducive to a reduction in surface roughness. The magnetic abrasives form a magnetic abrasive brush under the action of the external magnetic field, and the trajectory of the magnetic brush in the tube is a spiral line; the tube rotation and the magnet feed velocity have a great influence on the state of the spiral line, i.e., when using too large a tube rotation and magnet feed velocity, the state of the magnetic abrasives in the tube is chaotic and cannot form regular material removal. Too slow a magnetic feed velocity and too low a tube rotation velocity forms denser spirals with longer abrasive paths and tends to have excessive material removal. Therefore, moderate material removal and favorable surface defect layer removal should be selected.

#### 5.3.2. Interaction between Tube Rotation Velocity and MAPS Filling Quantity

Figure 9 shows the contour and three-dimensional response surface of the interaction between tube rotation velocity and magnet feed velocity on surface roughness for a magnet feed velocity of 100 mm/min. It can be seen from the figure that lower surface roughness can be obtained by using a moderate tube rotation velocity and more MAPS filling quantity. Less MAPs forms a more rigid magnetic abrasive brush, which tends to cause a scratching phenomenon on the inner wall of the tube, while the processing time is longer, and too little magnetic abrasive tends to lead to poor finishing results later. Therefore, if the magnetic abrasive will not produce clogging, more MAPs and moderate velocity should be chosen.

#### 5.3.3. Interaction between Magnetic Feed Velocity and MAPs Filling Quantity

Figure 10 shows the contour and three-dimensional response surface of the interaction between the magnet feed velocity and MAPs filling quantity on the surface roughness for a magnet feed velocity of 100 mm/min. From the figure, it can be seen that a moderate magnet feed velocity and more MAPs filling quantiy should be selected. From the figure, it can be seen that the surface roughness tends to decrease and then increase with the increase in the magnet feed velocity. When the magnet feed velocity increases, the feed velocity of some MAPs in the tube is smaller than the magnet feed velocity, and some MAPs are not involved in the finishding process, resulting in fewer MAPs involved in finishing, and the hysteresis phenomenon of MAPs is easy to produce, which leads to lower processing efficiency and incomplete removal of the defective layer.

### 5.4. Parameter Optimization and Experimental Verification

The optimization of processing parameters was then carried out. According to the prediction model, the optimal combination of process parameters was found based on the response value of minimizing the surface roughness of the inner wall of the nickel–titanium alloy cardiovascular support tube after machining. To verify the accuracy of the response surface method for the machining parameters, a validation test was conducted using this parameter. The surface roughness *Ra* was measured again using a 3D digital microscope, and the test results are shown in Table 7; a material removal thickness of only 5.2 μm does not affect the mechanical strength of the material. The SEM diagram with the optimized process parameters is shown in Figure 11. The error between the actual and predicted surface roughness values was 2.2%, and the optimal process parameters for processing the inner wall of the Ni-Ti alloy vascular stent tubing were obtained, which is of great significance for practical production.

## 6. Conclusions

CBN MAPs were prepared by plasma molten metal powder bonding with hard abrasives. It is ideally spherical in shape, with sharp cutting edges kept exposed to facilitate the introduction of MAF inside the ultralong cardiovascular stent tubing, and also has high cutting capacity; MAF device was developed to solve the problem of the removal of the inner wall defect layer of the ultra-long cardiovascular stent tubing. The regression model of the tube rotation velocity, magnet feed velocity, and MAPs filling quantity on the surface roughness was established by the response surface method, and the results of the residual and ANOVA analysis proved that it was a good fit. The interaction analysis shows that the influencing factors of surface roughness are tube rotation velocity > magnet feed velocity > MAPs filling quantity. The optimum process parameters were obtained as follows: tube rotation velocity 664 rpm, magnet feed velocity 89 mm/min, and MAP filling quantity 0.2 g. The surface roughness *Ra* of the tube inner wall was reduced from 0.356 μm to 0.083 μm by processing under the optimum process parameters. The problem of finishing the inner wall of the ultra slender nitinol vascular stent tubing was solved, which is important for practical applications. The preparation of high-performance spherical magnetic abrasives can better facilitate the application of MAF in the removal of defective layers from the inner wall of ultralong cardiovascular stents tube.

## Figures and Tables

**Figure 1 materials-16-03022-f001:**
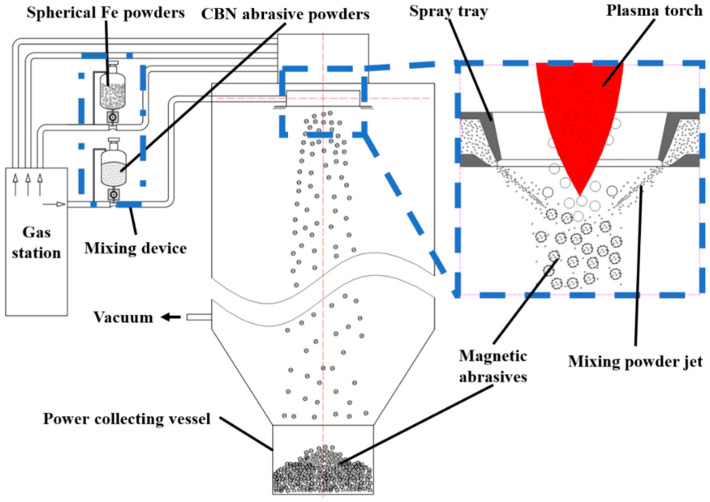
Principle diagram of magnetic abrasive preparation by plasma molten metal powder bonding with hard abrasives.

**Figure 2 materials-16-03022-f002:**
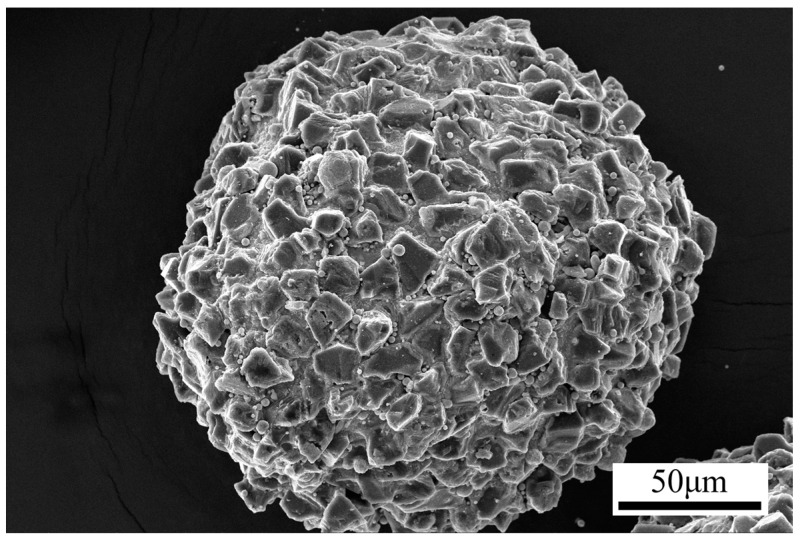
SEM image of CBN magnetic abrasive.

**Figure 3 materials-16-03022-f003:**
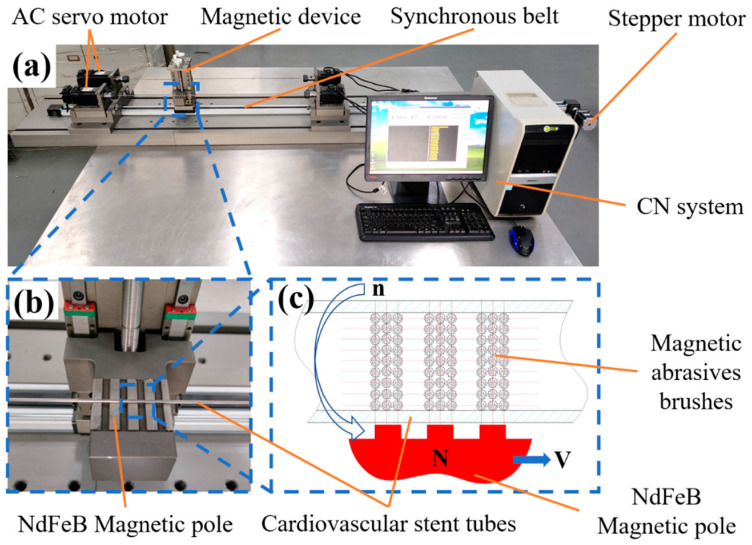
(**a**) MAF equipment for removing defective layers from the inner wall of ultralong cardiovascular stents tube; (**b**) magnetic device; (**c**) processing principle.

**Figure 4 materials-16-03022-f004:**
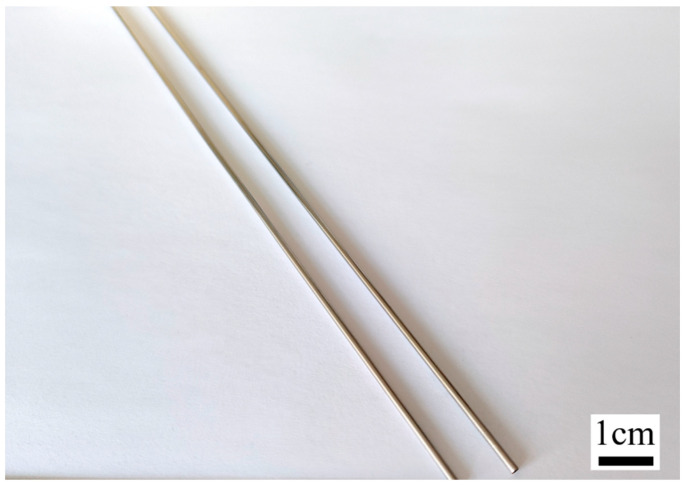
Ultraslender nickel–titanium alloy cardiovascular stents tube.

**Figure 5 materials-16-03022-f005:**
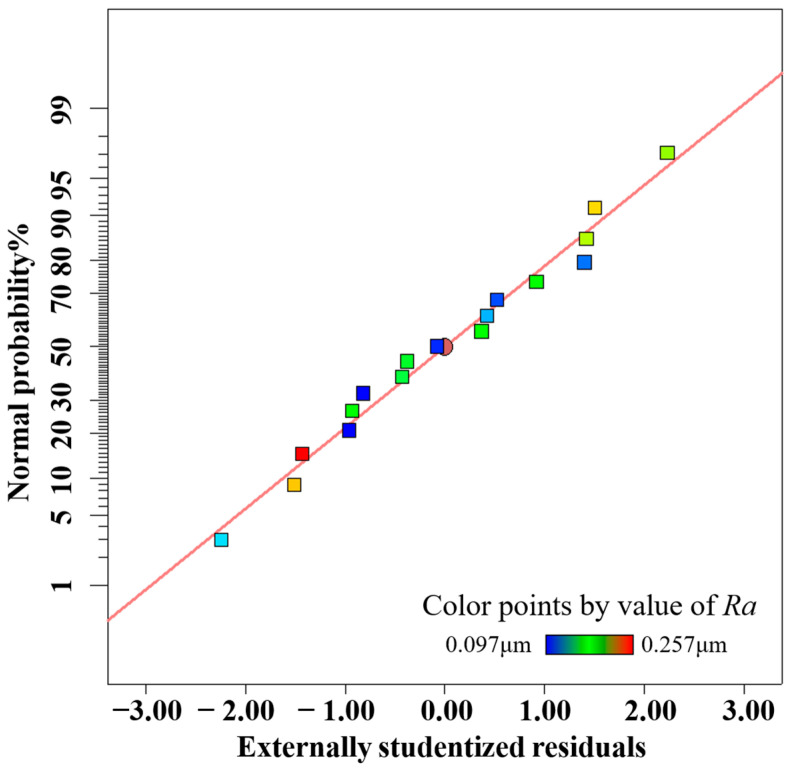
Normal plot of residuals.

**Figure 6 materials-16-03022-f006:**
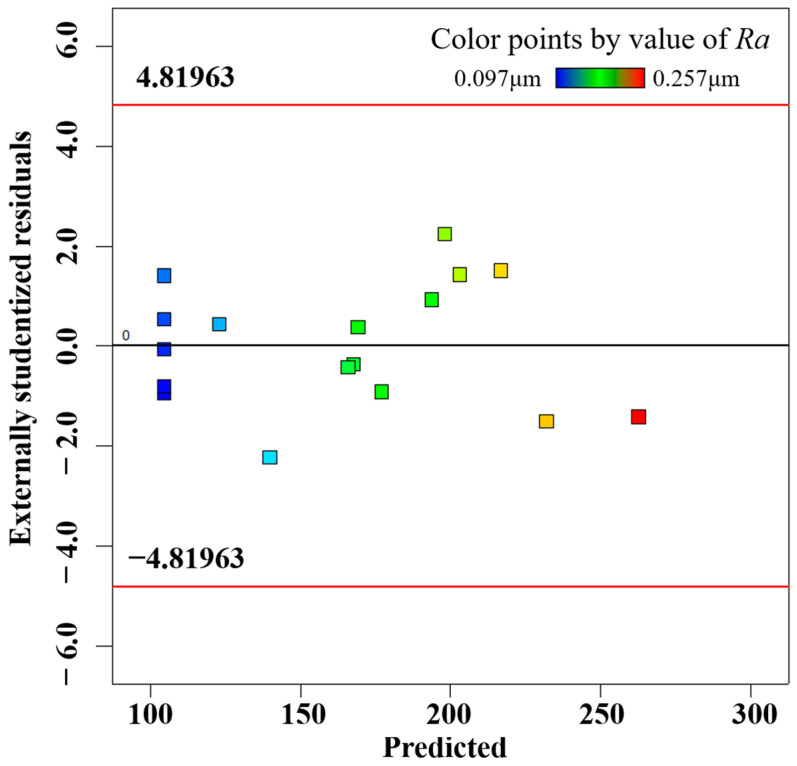
Residuals versus predicted plot.

**Figure 7 materials-16-03022-f007:**
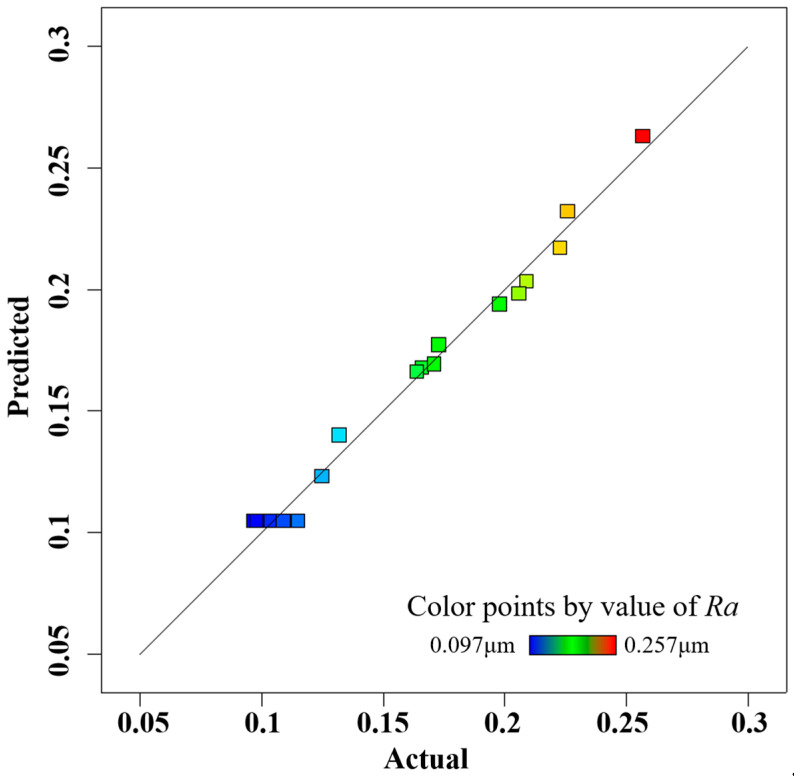
Plot of predicted versus actual values.

**Figure 8 materials-16-03022-f008:**
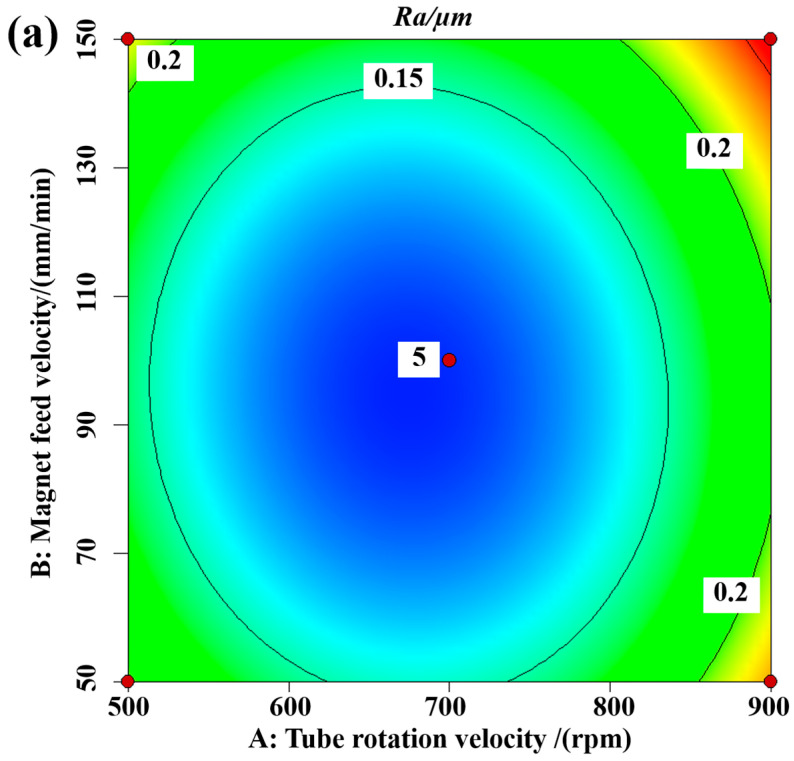
Interaction of tube rotation velocity and magnet feed velocity on surface roughness: (**a**) contour; (**b**) 3D response surface.

**Figure 9 materials-16-03022-f009:**
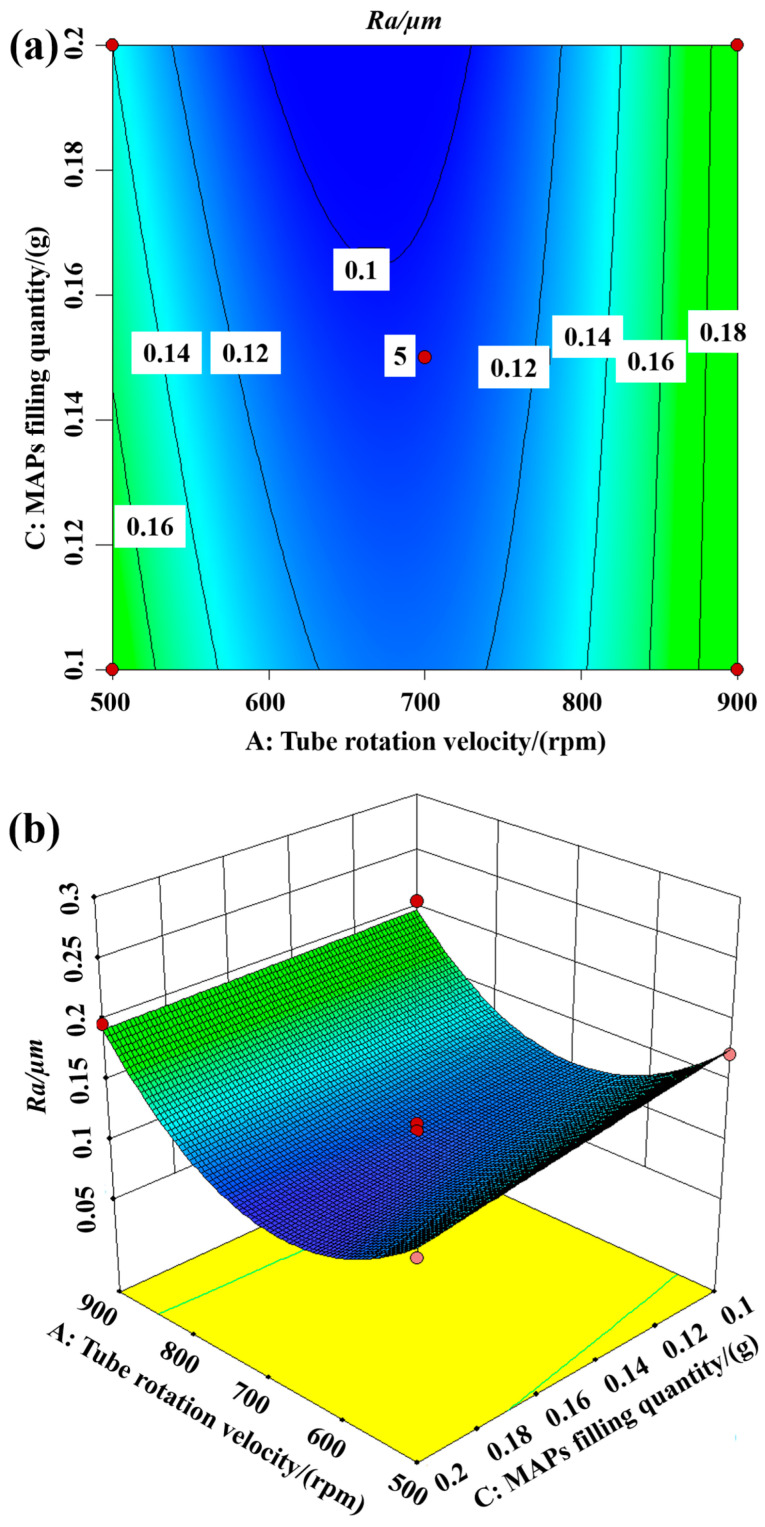
Interaction of tube rotation velocity and MAPs filling quantity on surface roughness: (**a**) contour; (**b**) 3D response surface.

**Figure 10 materials-16-03022-f010:**
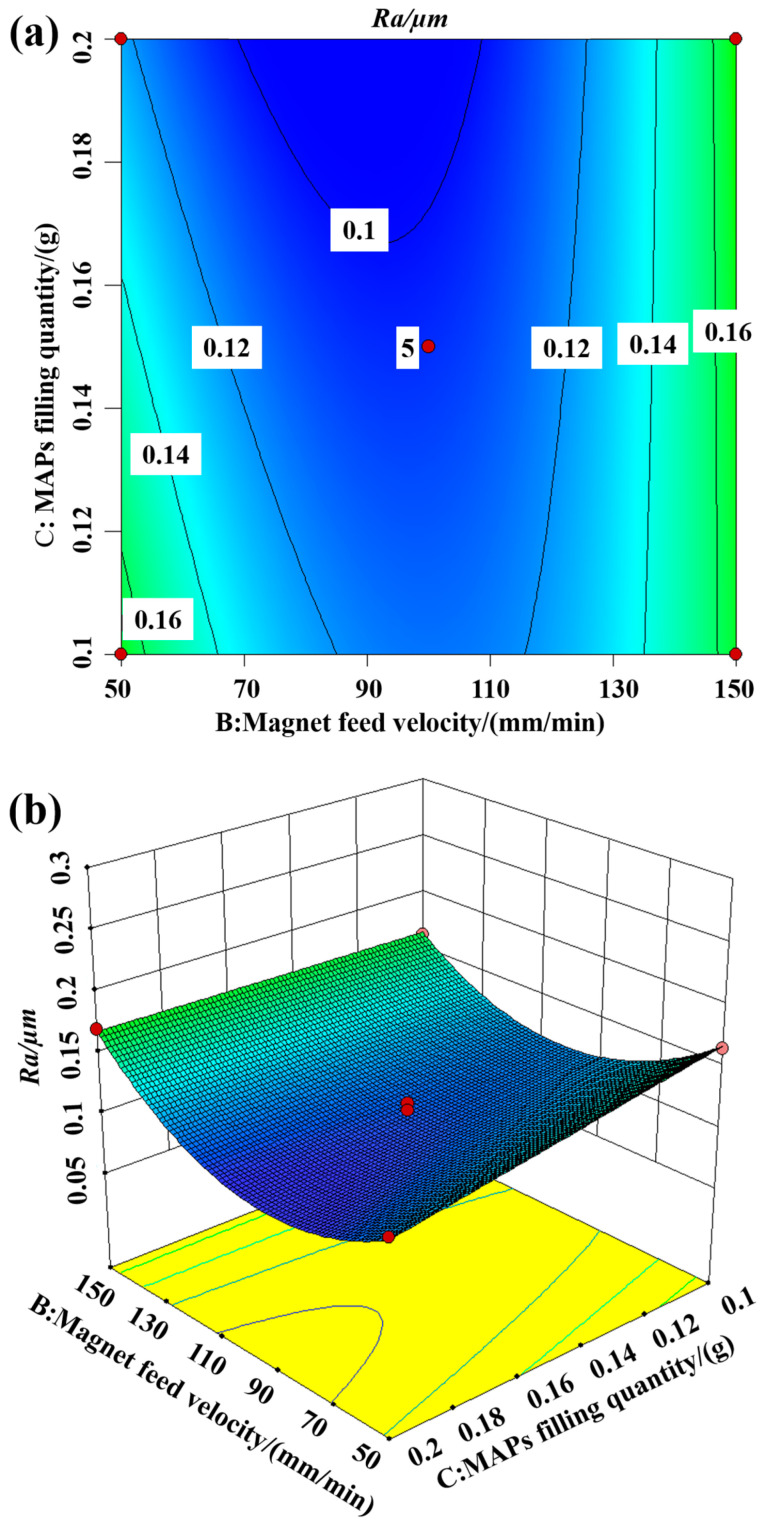
Interaction of magnet feed velocity and MAPs filling quantity on surface roughness: (**a**) contour plot; (**b**) 3D response surface plot.

**Figure 11 materials-16-03022-f011:**
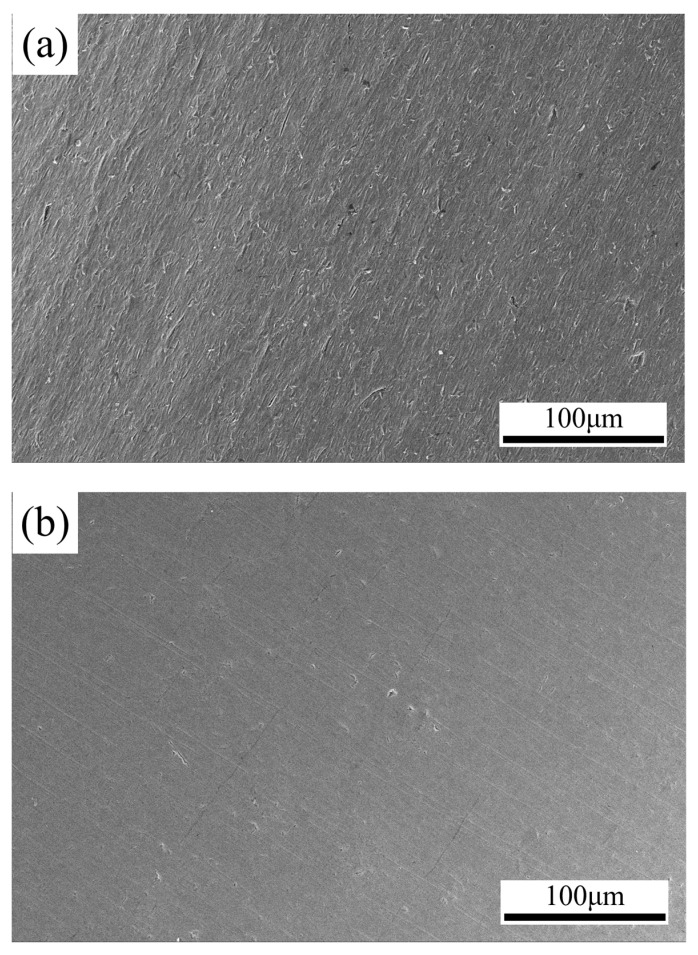
SEM surface morphology of the inner wall of the Ni-Ti alloy vascular stent tubing (**a**) before processing (0.356 μm); (**b**) after processing (0.083 μm).

**Table 1 materials-16-03022-t001:** Operating conditions of experiments.

Nozzle	Ring Seam
Nozzle cone angle (°)	65
Nozzle annular seam diameter (mm)	3.5
Nozzle bore diameter (mm)	46
Inlet pressure of nozzle (MPa)	0.8
Distance between nozzle and plasma generator (mm)	70
I (A)	700
Ar (L/h)	1000
H_2_ (L/h)	15
Iron powder (g/min)	40
CBN powder (g/min)	240
Equipment power (kW)	24.34

**Table 2 materials-16-03022-t002:** The chemical constituents of Ni-Ti alloy cardiovascular stent tube.

Element	Ni	Ti	O	N	K	C	Si	Al
Value	50~52	45~50	2~5	≤1	≤0.5	≤0.01	≤0.01	≤0.01

**Table 3 materials-16-03022-t003:** The performance parameters of Ni-Ti alloy.

Indicators	Density/g·cm^−3^	Modulus of Elasticity/GPa	Tensile Strength /MPa	Hardness (HRC)	Elongation/%	Shape Memory Recovery Rate/%	Fatigue Limit/MPa
Value	6.48	28~41	800~1500	30~40	1~20	98	100~800

**Table 4 materials-16-03022-t004:** Correspondence table of factor level and coding value.

Experiment Number	Tube Rotation Velocity *n*/(rpm)	Magnet Feed Velocity *v*/(mm/min)	MAPs Filling Quantity *δ*/g
−1	500	50	0.1
0	700	100	0.15
1	900	150	0.2
Radius of change	200	50	0.05

**Table 5 materials-16-03022-t005:** Test plan and results.

Experiment Number	Tube Rotation Speed *n*/(rpm)	Magnet Feed Velocity *v*/(mm/min)	MAP Filling Quantity *δ*/g	Surface Roughness *Ra*/μm
1	700	150	0.1	0.164
2	700	150	0.2	0.171
3	900	100	0.1	0.206
4	700	100	0.15	0.097
5	700	100	0.15	0.109
6	700	100	0.15	0.115
7	700	50	0.1	0.166
8	500	150	0.15	0.223
9	700	50	0.2	0.125
10	700	100	0.15	0.104
11	900	100	0.2	0.198
12	900	50	0.15	0.226
13	900	150	0.15	0.257
14	700	100	0.15	0.098
15	500	50	0.15	0.209
16	500	100	0.1	0.173
17	500	100	0.2	0.132

**Table 6 materials-16-03022-t006:** ANOVA results of surface roughness.

Source	Sum of Squares	df	Mean Square	F	Prob > F	
Model	4.1 × 10^−2^	9	4.546 × 10^−3^	58.34	<0.0001	significant
*A*—Pipe Rotation Speed	2.813 × 10^−3^	1	2.813 × 10^−3^	36.09	0.0005	
*B*—Magnetic Pole Feed Rate	9.901 × 10^−4^	1	9.901 × 10^−4^	12.71	0.0092	
*C*—Magnetic Abrasive Filling Quantity	8.611 × 10^−4^	1	8.611 × 10^−4^	11.05	0.0127	
*AB*	7.225 × 10^−5^	1	7.225 × 10^−5^	0.93	0.3677	
*AC*	2.723 × 10^−4^	1	2.723 × 10^−4^	3.49	0.1038	
*BC*	5.760 × 10^−4^	1	5.760 × 10^−4^	7.39	0.0298	
*A* ^2^	0.022	1	0.022	283.63	<0.0001	
*B* ^2^	0.011	1	0.011	144.43	<0.0001	
*C* ^2^	1.684 × 10^−7^	1	1.684 × 10^−7^	2.161 × 10^−3^	0.9642	
Residual	5.454 × 10^−4^	7	7.792 × 10^−5^			
Lack of Fit	3.162 × 10^−4^	3	1.054 × 10^−4^	1.84	0.2802	not significant
Pure Error	2.292 × 10^−4^	4	5.730 × 10^−5^			
Cor Total	0.041	16				
*R*-Squared = 0.9868				*Adj R*-Squared = 0.9699		

**Table 7 materials-16-03022-t007:** Experimental results after parameter optimization.

Experimental 1	Experimental 2	Experimental 3	Mean Experimental	Predictive	Error
0.095 μm	0.095 μm	0.083 μm	0.093 μm	0.089 μm	4.3%

## Data Availability

Not applicable.

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
