# Peer review of "Investigation of MAF for Finishing the Inner Wall of Super-Slim Cardiovascular Stents Tube"

_materials, 2023, doi:10.3390/ma16083022_

Round 1

Reviewer 1 Report

In this study, magnetic particle grinding was used to overcome the difficulty of removing the defect layer from the inner wall of a very slim cardiovascular stent tube. First, a new technique called plasma molten metal powder bonding with hard abrasives was used to create a spherical CBN magnetic abrasive. Next, a magnetic particle grinding device was created to remove the defect layer from the inner wall of an ultra-fine, long cardiovascular stent tube. The findings demonstrate that the prepared spherical CBN magnetic abrasive is perfectly spherical, has sharp cutting edges covering the iron matrix's superficial layer, and that the magnetic particle grinding apparatus developed for use with ultra-fine, long cardiovascular stent tubing satisfies the processing requirements.

I am not a specialist in the field of engineering (materials engineering), but the paper used some statistical methods with great skill, and presented convincing and satisfactory results to a large extent.

Accordingly, it was proposed to accept the paper after making some amendments and improvements in the following aspects:

1-English should be revised well, many typos are spotted, see for example L11 in the abstract "from The roughness" should be "from the roughness".

2-It is not recommended to include numerical results in the abstract.

3-Add a list for all abbreviations.

4-Regarding Eq 1: please add its reference(s) and the related papers which applied it, and why this model is chosen, you should motivate this point very carefully.

5-Adjust the font of Eq2.

6-All figures need to be improved, see example: 5,6,7 and 11.

7-How can readers and those interested in the science of materials engineering extend and generalize this work, the reader needs some future points to work on.

8-Finally: It may be appropriate to use more than one statistical model and make some comparisons to choose the best models. In fact, this paper used only one model, and based on this model, the whole paper was built. Is it possible to present some results using another competing model and make some comparisons? If the authors can do that, the paper will be much better. 

Author Response

Dear Reviewer:

I would first like to thank you for your letter and the reviewer’s comments concerning our manuscript entitled “Investigation of MAF to remove defective layers on the inner wall of super   slim cardiovascular stents tube” The comments you made were all valuable and helpful for revising and improving our paper and the important guiding significance to our research. We have substantially revised our manuscript after reading the comments provided by the reviewer. I hope to be met with approval.

Revised portions are marked in red throughout the paper, and the main corrections in the paper and the responses to the reviewer’s comments are as follows:

Responds to reviewer’s comments:

Reviewer 1:

1) English should be revised well, many typos are spotted, see for example L11 in the abstract "from The roughness" should be "from the roughness".

Answer: We are sorry for our negligence. I have revised my manuscript, and the specific revised parts have been marked in red in the submitted manuscript.

2) It is not recommended to include numerical results in the abstract.

Answer: Thank you for your comments on this section. As this paper is doing a parametric optimisation study, we think it is very important that the optimised numbers appear in the abstract and that their presence facilitates the reader's reading.

3) Add a list for all abbreviations.

Answer: We are sorry for our negligence. I have revised my manuscript, and the specific revised parts have been marked in red in the submitted manuscript.

4) Regarding Eq 1: please add its reference(s) and the related papers which applied it, and why this model is chosen, you should motivate this point very carefully.

Answer: We are sorry for our negligence. I have revised my manuscript, and the specific revised parts have been marked in red in the submitted manuscript.

5) Adjust the font of Eq2.

Answer: We are sorry for our negligence. I have revised my manuscript, and the specific revised parts have been marked in red in the submitted manuscript.

6) All figures need to be improved, see example: 5,6,7 and 11.

Answer: We are sorry for our negligence. I have revised my manuscript, and the specific revised parts have been marked in red in the submitted manuscript.

7) How can readers and those interested in the science of materials engineering extend and generalize this work, the reader needs some future points to work on.

Answer: Thank you for your comments on this section. I have revised my manuscript, and the specific revised parts have been marked in red in the submitted manuscript.

8) Finally: It may be appropriate to use more than one statistical model and make some comparisons to choose the best models. In fact, this paper used only one model, and based on this model, the whole paper was built. Is it possible to present some results using another competing model and make some comparisons? If the authors can do that, the paper will be much better.

Answer: Thank you for your comments on this section. Although only one model was used in this paper, the model developed was validated with a model prediction error of only 4.3% and a reduction in roughness (Ra) from 0.356 μm to 0.083 μm with accurate predictions from the prediction model. A study on the use of an alternative model for comparison will be carried out in the next paper and in time to " Materials" in time for submission to the journal.

We tried our best to improve the revised manuscript and made some changes in the revised manuscript. And here we did not list the changes but marked in red in the revised manuscript, using the "Track Changes" function in Microsoft Word.

We appreciate Editors/Reviewer’s warm work earnestly, and hope that the correction will meet with approval. Once more, thank you very much for your comments and suggestions.

Thanks very much again for your attention to our manuscript. Once again, thank you for your help to our manuscript processing.

Yours sincerely,

Yugang Zhao

Reviewer 2 Report

the article was prepared correctly. Many modern and adequate devices were used to verify the presented theses. The article touches on a topic important for practitioners who want to maintain the patency of stents in the long term.

It is puzzling why titanium oxide on the surface matters inside the stent - after all, there is no contact with any tissues. In my opinion, this feature is irrelevant in this study.

The literature is extensive, there are many relevant pictures and tables.

In my opinion, the article can be published.

Author Response

Dear Reviewer:

I would first like to thank you for your letter and the reviewer’s comments concerning our manuscript entitled “Investigation of MAF to remove defective layers on the inner wall of super   slim cardiovascular stents tube” The comments you made were all valuable and helpful for revising and improving our paper and the important guiding significance to our research. We have substantially revised our manuscript after reading the comments provided by the reviewer. I hope to be met with approval.

Revised portions are marked in red throughout the paper, and the main corrections in the paper and the responses to the reviewer’s comments are as follows:

Responds to reviewer’s comments:

Reviewer 2:

1) It is puzzling why titanium oxide on the surface matters inside the stent - after all, there is no contact with any tissues. In my opinion, this feature is irrelevant in this study.

Answer: Thank you for your comments on this section. Ni-Ti cardiovascular stents have developed defects such as pits, bulges, and cracks in the inner wall during the drawing production process. The presence of these defects puts the patient's health at risk, and the inner wall of cardiovascular stent tubing must have a low roughness, which is important to improve the fatigue resistance and cytocompatibility of medical implants. It is mentioned in the following literature.

[1] Valentina Finazzi, Ali Gökhan Demir, Carlo Alberto Biffi, Francesco Migliavacca, Lorenza Petrini, Barbara Previtali, Design and functional testing of a novel balloon-expandable cardiovascular stent in CoCr alloy produced by selective laser melting, Journal of Manufacturing Processes, 2020(55), 161-173, https://doi.org/10.1016/j.jmapro.2020.03.060

[2] Kotousov, A., Bortolan Neto, L. & Rahman, S.S. Theoretical model for roughness induced opening of cracks subjected to compression and shear loading. Int J Fract 172, 9–18 (2011). https://doi.org/10.1007/s10704-011-9642-6

[3] Brecht Van Hooreweder, Karel Lietaert, Bram Neirinck, Nicholas Lippiatt, Martine Wevers, CoCr F75 scaffolds produced by additive manufacturing: Influence of chemical etching on powder removal and mechanical performance, Journal of the Mechanical Behavior of Biomedical Materials, 2017(70):60-67, https://doi.org/10.1016/j.jmbbm.2017.03.017

We tried our best to improve the revised manuscript and made some changes in the revised manuscript. And here we did not list the changes but marked in red in the revised manuscript, using the "Track Changes" function in Microsoft Word.

We appreciate Editors/Reviewer’s warm work earnestly, and hope that the correction will meet with approval. Once more, thank you very much for your comments and suggestions.

Thanks very much again for your attention to our manuscript. Once again, thank you for your help to our manuscript processing.

Yours sincerely,

Yugang Zhao

Reviewer 3 Report

The abstract does not explain the significance of reducing the roughness of the inner wall of the stent tubing. The review lacks a critical evaluation of previous studies on the internal polishing of Ni-Ti alloy cardiovascular stent tubing. It does not explain how MAF works or the parameters that need to be controlled to achieve the desired surface finish.

The study only focused on surface roughness and did not consider other important factors such as material removal rate, tool wear, or processing time. The underlying physical or chemical processes involved in magnetic abrasive finishing are not thoroughly discussed, limiting the applicability of the results and preventing a deeper understanding of the phenomenon.

The presented validation test measures surface roughness. It does not evaluate the overall quality or performance of the finished product. It is possible that optimizing the process for surface roughness may compromise other aspects of the cardiovascular support tube.

Author Response

Dear Reviewer:

I would first like to thank you for your letter and the reviewer’s comments concerning our manuscript entitled “Investigation of MAF to remove defective layers on the inner wall of super   slim cardiovascular stents tube” The comments you made were all valuable and helpful for revising and improving our paper and the important guiding significance to our research. We have substantially revised our manuscript after reading the comments provided by the reviewer. I hope to be met with approval.

Revised portions are marked in red throughout the paper, and the main corrections in the paper and the responses to the reviewer’s comments are as follows:

Responds to reviewer’s comments:

Reviewer 3:

1) The abstract does not explain the significance of reducing the roughness of the inner wall of the stent tubing.

Answer: We are sorry for our negligence. I have revised my manuscript, and the specific revised parts have been marked in red in the submitted manuscript.

2) The review lacks a critical evaluation of previous studies on the internal polishing of Ni-Ti alloy cardiovascular stent tubing.

Answer: We are sorry for our negligence. I have revised my manuscript, and the specific revised parts have been marked in red in the submitted manuscript.

It does not explain how MAF works or the parameters that need to be controlled to achieve the desired surface finish.

Answer: Fig. 3(c) shows the principle of magnetic abrasive processing of the inner wall of a Ni-Ti alloy vascular stent tube. The iron-based CBN magnetic abrasive is added to the inside of the Ni-Ti vascular stent tubing, and under the action of the external magnetic field, the magnetic abrasive grains are magnetized by the external magnetic field to form a magnetic grain brush with certain cutting ability and rigidity. The nitinol vascular stent tubing is clamped by two end collets, which are mounted on servo motors at both ends. When the servo motor drives the tube to rotate and the magnetic pole moves reciprocally in the direction of the axis, the magnetic grain brush moves relative to the inner wall surface of the tube, and the magnetic abrasive, which is magnetized to form the "abrasive brush", moves spirally on the surface of the workpiece to produce the effect of sliding, cutting and plowing on the inner wall of the tube, and then the inner wall is magnetically ground.

3) The study only focused on surface roughness and did not consider other important factors such as material removal rate, tool wear, or processing time. The underlying physical or chemical processes involved in magnetic abrasive finishing are not thoroughly discussed, limiting the applicability of the results and preventing a deeper understanding of the phenomenon.

Answer: Thank you for your comments on this part. After careful consideration, the study of the "material removal rate, tool wear, or processing time" will be examined in the next paper.

4) The presented validation test measures surface roughness. It does not evaluate the overall quality or performance of the finished product. It is possible that optimizing the process for surface roughness may compromise other aspects of the cardiovascular support tube.

Answer: Thank you for your comments on this part. I have revised my manuscript, and the specific revised parts have been marked in red in the submitted manuscript.

We tried our best to improve the revised manuscript and made some changes in the revised manuscript. And here we did not list the changes but marked in red in the revised manuscript, using the "Track Changes" function in Microsoft Word.

We appreciate Editors/Reviewer’s warm work earnestly, and hope that the correction will meet with approval. Once more, thank you very much for your comments and suggestions.

Thanks very much again for your attention to our manuscript. Once again, thank you for your help to our manuscript processing.

Yours sincerely,

Yugang Zhao
